# Identification of *Bacillus velezensis* SBB and Its Antifungal Effects against *Verticillium dahliae*

**DOI:** 10.3390/jof8101021

**Published:** 2022-09-28

**Authors:** Wei-Yu Wang, Wei-Liang Kong, Yang-Chun-Zi Liao, Li-Hua Zhu

**Affiliations:** 1College of Forestry, Nanjing Forestry University, Nanjing 210037, China; 2College of Plant Protection, China Agricultural University, Beijing 100193, China; 3Co-Innovation Center for Sustainable Forestry in Southern China, Nanjing 210037, China

**Keywords:** *Verticillium dahliae*, *Bacillus velezensis*, biological control, antagonistic effect

## Abstract

Traditional control methods have drawbacks in controlling Verticillium wilt diseases caused by *Verticillium dahliae* Kleb.; therefore, an efficient and environmentally friendly strategy for disease control must be identified and the mechanisms determined. In this study, a soil-isolated strain SBB was identified as *Bacillus velezensis* based on 16S rRNA, *gyrA*, and *gyrB* gene sequences. In vitro, strain SBB had excellent inhibitory effects on *V. dahliae*, with the highest inhibition rate of 70.94%. Moreover, strain SBB inhibited production of the conidia of *V. dahliae* and suppressed the production of microsclerotia and melanin. Through gas chromatograph–mass spectrometer analysis, nine compounds were detected from the volatile organic compounds produced by SBB, among which 2-nonanol, 2-heptanone, 6-methyl-2-heptanone, and 2-nonanone could completely inhibit *V. dahliae* growth. Strain SBB produced cellulase, amylase, protease, and siderophore. During inhibitory action on *V. dahliae*, strain SBB showed upregulated expression of genes encoding non-volatile inhibitory metabolites, including difficidin, bacilysin, and bacillaene, at 1.923-, 1.848-, and 1.448-fold higher, respectively. Thus, our study proved that strain SBB had an efficient antagonistic effect on *V. dahliae*, suggesting strain SBB can be used as a potential biological control agent against Verticillium wilt.

## 1. Introduction

*Verticillium dahliae* Kleb. is a serious soilborne pathogenic fungus that can infect plants and cause Verticillium wilt, causing a series of symptoms such as leaf yellowing, wilting, necrosis, curling, and plant death in host plants. *V. dahliae* invades plants from the roots in the soil and colonizes the vascular bundles. *V. dahliae* has been reported to produce phytotoxins and cell-wall-degrading enzymes to harm hosts. As the major inoculum source and primary long-term survival structures, microsclerotia and melanized mycelium of *V. dahliae* play an important role in the disease cycle [1,2,3,4]. This pathogen is distributed all over the world and can infect more than 660 types of plants, including cotton (*Gossypium* spp.), eggplant (*Solanum melongena* L.), potato (*Solanum tuberosum* L.), and olive (*Canarium album* (Lour.) Rauesch.), causing huge ecological and economic losses to agricultural and forestry production [5,6,7,8].

Due to the high stress resistance, long lifespan, and strong drug resistance of the microsclerotia, *V. dahliae* is difficult to control using traditional methods. Chemicals usually work with high dosages, which sometimes may cause a phytotoxic effect. The disease control effect of soil fumigation is not stable, and some fumigants, such as methyl bromide, which is an ozone-depleting substance, may harm the environment [9,10,11]. The optimal planting mode used to control Verticillium wilt is only effective when planting density is low, and disease-resistant breeding is also limited by the long breeding cycle and the lack of genetic resources [1,12,13].

The progress of sustainable agriculture and environmental protection make biological control an attractive alternative to protect plants from Verticillium wilt. In recent years, *Bacillus* spp. have received extensive attention because of their ability to produce a variety of antifungal active substances and to exist in the soil with strong vitality and high colonization rates [14,15,16]. *Bacillus* usually have a variety of biological control mechanisms, such as competing with pathogenic fungi for space and nutrient sites, inducing plant system resistance (ISR), and producing a variety of hydrolases [17,18,19,20]. For example, *B. amyloliquefaciens* S170 and *B. pumilus* S9 isolated from soil showed a good inhibitory effect on rice blast caused by *Magnaporthe oryzae* [21]. *Bacillus subtilis* BY-2 was proven effective in the control of *Sclerotinia sclerotiorum* on oilseed rape [22]. Han et al. demonstrated that *B. velezensis* FZB42 could efficiently antagonize *Phytophthora sojae* and control soybean root rot [23]. *B. velezensis* K165 could trigger ISR in eggplant against *V. dahliae* [24].

In this study, the *Bacillus* strain SBB, previously isolated from soil, was the research object, and its specific taxonomic status was first identified using molecular biology. Because the specific antagonistic effect of strain SBB on *V. dahliae* was unknown, we investigated its antagonism effect on *V. dahliae*, including mycelial growth, spore production, and microsclerotium formation. The non-volatile substances and volatile organic compounds (VOCs) produced by SBB were identified. The results of this study provided new possibilities for the biological control of Verticillium wilt caused by *V. dahliae* and had important theoretical significance for the development and utilization of related biopesticides.

## 2. Materials and Methods

### 2.1. Bacterial and Fungal Strains

The strain SBB used in this study was previously isolated from soil at Beidashan, near Nanjing Forestry University (118°46.615′ E, 31°43.195′ N). It was routinely cultured on Luria–Bertani (LB) medium at 28 °C. The target pathogen *V. dahliae* At13, isolated from *Acer truncatum* Bunge in Shandong Province, China [25], was maintained on potato dextrose agar medium (PDA; Difco-Becton, Dickinson, Sparks, MD, USA) at 25 °C.

### 2.2. Identification of Strain SBB

To identify SBB, its 16S rRNA, *gyrA*, and *gyrB* gene sequences were analyzed. The strain was inoculated into LB medium and shaken overnight at 30 °C. The genomic DNA was extracted using a modified freeze–thawing method [26]. Bacterial universal primers 27F (5′-AGAGTTTGATCCTGGCTCAG-3′) and 1492R (5′-CGGCTACCTTGTTACGAC-3′) were used to amplify partial sequences of 16S rRNA [27]. The *gyrB* gene was amplified with specific primers UP1S (5′-GAAGTCATCATGACCGTTCTGC-3′) and UP2rS (5′-AGCAGGGTACGGATGTGCGAGCC-3′) [28], and the *gyrA* gene was amplified with specific primers 7237F (5′-CAGTCAGGAAATGCGTACGTCCTT-3′) and 8261F (5′-CAAGGTAATGCTCCAGGCATTGCT-3′) [29]. The thermal circulation conditions for 16S rRNA and *gyrB* were as follow: 94 °C for 5 min; 30 cycles at 94 °C for 30 s, 56 °C for 35 s, and 72 °C for 90 s; and a final extension at 72 °C for 10 min. The thermocycling procedure for *gyrA* involved initial denaturation at 94 °C for 5 min, followed by 30 cycles at 94 °C for 30 s, 58 °C for 45 s, and 72 °C for 90 s, and a final extension at 72 °C for 10 min. The PCR products were tested on a 1% agarose gel, and the remaining PCR product was sent to Springen Biotechnology Co., Ltd. (Nanjing, China) for sequencing. The sequences were analyzed using BLAST (basic local alignment search tool) alongside NCBI data (https://www.ncbi.nlm.nih.gov/, accessed on 14 August 2022). The phylogenetic trees were constructed using MEGA7.0 [30], based on the neighbor-joining (NJ) method with 1000 bootstrap replications. The sequence data that support the findings of this study are openly available in the GenBank of NCBI under the accession numbers: OP218504 (16S rRNA), OP297674 (*gyrA*) and OP297675 (*gyrB*).

### 2.3. Determining the Antagonistic Effect of SBB on V. dahliae

#### 2.3.1. Effect of SBB on Mycelial Growth

The direct inhibitory effect of strain SBB on *V. dahlia**e* growth was determined by dual culture plate assay. A 6-mm diameter plug of *V. dahliae* was inoculated in the center of a Petri dish containing 20 mL of PDA, and 2 mL of SBB solution was dipped into the Petri dish and scribed symmetrically at 2.5 cm from the center. Each experiment was repeated three times and incubated at 25 °C for 10 d to observe the antagonistic effect.

#### 2.3.2. Effect of SBB VOCs on Mycelial Growth

The antifungal activity of strain SBB VOCs was assayed using a two sealed Petri dishes method [31]. The bottom plate of one Petri dish contained 20 mL LB and the bottom plate of the other Petri dish contained 20 mL of PDA. Of the SBB suspension, 100 μL was applied to LB medium and a 6-mm diameter agar plug of *V. dahliae* was placed in the center of the PDA plate. Then, the bottom plates of both Petri dishes were covered and sealed with Parafilm. Each experiment was repeated three times and incubated at 25 °C for 10 d to observe the antagonistic effect.

#### 2.3.3. Effect of SBB Fermentation Filtrate at Different Time Periods on Mycelial Growth

The activated SBB strain was inoculated into 25 mL of liquid LB medium at 1% inoculum and incubated at 28 °C and 200 rpm in a constant temperature shaker. The fermentation broth was collected after 24, 48, 72, 96, and 120 h of incubation and centrifuged at 8000× *g* for 20 min; the supernatant was filtered through a 0.22-μm microporous membrane to obtain the SBB fermentation filtrate of each time period.

The fermentation filtrate of the SBB strain collected from each time point was added at 20% in PDA medium, and PDA with 20% of LB medium was added as the control. After solidification, they were all inoculated with 6-mm diameter *V. dahliae* plugs in the center of the medium. Each experiment was repeated three times and incubated at 25 °C for 10 d to observe the antagonistic effect. Phytopathogenic fungal inhibition (%) = (Cd − Td) × 100%/Cd, where Cd is the colony diameter on the control PDA and Td is the colony diameter on the treated PDA.

### 2.4. Effect of SBB Filtrate on Spore Production

The collected 48-h fermentation filtrate of strain SBB was added and mixed with 25 mL of complete medium (CM) at 1%, 5%, 10%, and 20%, respectively, CM medium without 48-h fermentation filtrate of strain SBB was used as a control. Five 6-mm diameter *V. dahliae* plugs were inoculated to each medium and incubated at 25 °C with shaking at 150 rpm for 2 d. Each experiment was repeated three times. After the incubation, the spore fluid was collected by filtration through two layers of gauze, and the spores were counted under a microscope using a hemocytometer plate. Five grids (one in the center and four in the corners of the plate) were counted and averaged. Spore production = average number of spores per grid × 25 × 10^4^.

### 2.5. Effect of SBB Filtrate on Microsclerotia Formation

The 48-h fermentation filtrate of strain SBB was added and mixed with 20 mL of PDA medium at 1%, 5%, 10%, and 20%. After the plates were solidified, 6-mm diameter *V. dahliae* plugs were inoculated at the center of the medium. The plates were incubated at 25 °C for 14 d. Each experiment was repeated three times. After incubation, the plates were observed under a Zeiss microscope (SteREO Discovery V.20, Wetzlar, Germany), and the number of micronuclei per square millimeter was counted.

### 2.6. Effect of SBB Filtrate on Melanin Formation

Referring to the method of Bashyal et al. [32], after 10 d of dry culture, equal weights of mycelium samples were obtained from the untreated and strain SBB-treated *V. dahliae*. The mycelium was placed in a 50-mL centrifuge tube and 10 mL of distilled water was added. The solution was then boiled in a water bath for 5 min, centrifuged at 13,400× *g* for 5 min, washed, and centrifuged again; then, 10 mL of 1 M NaOH was added. The samples were heated in an autoclave at 120 °C for 20 min, cooled and adjusted to pH 2.0 for precipitation, dissolved in 1 M NaOH, centrifuged at 13,400× *g* for 15 min, and washed three times with distilled water. The extracted dried product was dissolved in 5 mL of 1 M NaOH and centrifuged at 13,400× *g* for 10 min, and the supernatant was transferred to a new tube. The OD value at 400 nm (OD400) was measured and used to calculate the melanin content: melanin content (g/L) = OD400 × 0.105 × N (N is the number of dilutions) [33]. The experiment was repeated twice, with three replicates for each treatment.

### 2.7. Detection of Antifungal Active Substance Genes in Strain SBB

Based on the previous research of common genes regulating the biosynthesis of antimicrobial active substances in *Bacillus* spp., and using the total DNA of strain SBB as a template, PCR amplification was performed for nine antimicrobial substances possibly contained in strain SBB (surfactin, fengycin, iturin, bacillomycin, mycosubtilin, bacilysin, bacillaene, difficidin, and bacillibactin) using the primer sequences shown in Table 1 [34,35,36,37,38].

### 2.8. Extracellular Enzyme and Siderophore Activities

The activity of strain SBB in secreting extracellular enzymes was tested by plate-based assays. Strain SBB was incubated on PDA plates containing amylase, cellulase, protease, β-1,3-glucanase, chitinase, and phosphodiesterase, with each group repeated three times before being incubated at 28 °C for 3 d [39,40]. The cellulase assay media was flooded with 0.01% Congo red solution for 30 min and then observed for any ring-like transparent circle produced, which would indicate that SBB produced cellulase. The amylase assay media was flooded with iodine solution for 30 min and examined for any production of a ring-shaped transparent circle, which would indicate that SBB produced amylase. Another four enzyme assay media were used to determine whether strain SBB produced the corresponding extracellular enzymes by directly observing the presence or absence of ring-shaped transparent circles.

Strain SBB was incubated in Salmonella Arizona Agar medium for 48 h, then washed twice with 1 × phosphate buffer saline (pH 7.5) and filtered with 0.22-μm microporous membrane to remove bacteria; the filtrate was then collected. Then, 50 μL of fermentation filtrate was added to each well, made using a punch in a Chrome Azurol S (CAS) plate, incubated at 28 °C for 2 d, and examined for any production of a yellow halo [41].

### 2.9. Gas Chromatograph-Mass Spectrometer (GC-MS) Analysis of VOCs in SBB and Effect of Commercial VOCs on Fungal Growth

Single colony SBB strains were added to 250-mL conical flasks containing 100 mL of LB liquid medium and incubated for 2 d in a constant temperature shaker at 28 °C and 200 rpm. An equal amount of LB liquid medium without SBB inoculation was used as a control. To prevent the escape of VOCs, the conical flasks were sealed with tin foil. For this experiment, 65-μm polydimethylsiloxane/divinylbenzene fiber tips were used to determine the volatile gases produced by strain SBB. The fiber tips were aged before first use. In this experiment, pretreatment temperature of the fiber tip was 250 °C and pretreatment time was 30 min [42]. The cultured bacterial samples were shaken and placed in a water bath at 40 °C. An aged solid-phase microextraction fiber head was inserted into the tin foil. The samples were extracted by sorptive extraction for 30 min. After extraction, the fiber head was quickly retracted, and the needle removed and immediately inserted into the gasification chamber of the GC. The fiber head was extruded, and the target was thermally cracked for 3 min using the high temperature of the gasification chamber.

The GC-MS conditions were as follows: Rtx 5 quartz capillary column; helium as carrier gas; 230 °C inlet temperature; 40 °C starting temperature, maintained for 3 min, increased at 10 °C/min to 95 °C, then at 30 °C/min to 230 °C, and the 230 °C maintained for 5 min; ion source was EI source; electron energy was 70 eV; and the spectral search was performed using Nist 05 search and Nist 05 library [43]. Based on the identified gas composition, the corresponding pure standards were purchased to demonstrate their potential antifungal activity. All the standards were diluted to different concentrations (0.06, 0.3, 0.6, 1.3 µL/mL) and co-cultured with *V. dahliae* at 25 °C for 7 d to observe their antagonistic effects. Each experiment was repeated three times.

### 2.10. RNA Extraction and Real-Time Quantitative PCR (RT-qPCR) Analysis of SBB

A 6-mm diameter *V. dahliae* plug was inoculated at the center of a Petri dish containing 20 mL of PDA; 2 mL of SBB solution was dipped into the dish at 2.5 cm from the center and incubated in a constant temperature incubator at 25 °C for 7 d. SBB incubated in PDA medium without a *V. dahliae* plug was used as the control. The SBB was collected from different treatment groups after 7 d of incubation, and total RNA was extracted using the TRIzol method. After DNase I treatment, 2 μg of ribonucleic acid was added to the 20-μL reaction system, and first-strand cDNA was synthesized using the reverse transcription system. The cDNA samples were prepared using HiScript II Q Select Supermix for RT-qPCR (CAT: 11202ES08; Yeasen, Shanghai, China) using 1.0 µL of cDNA diluted 1:10 as the template [44].

The 11 antibiotic genes (*srfAD*, *fenD*, *ituC*, *ituD*, *bmyA*, *mycB*, *bacAB*, *bacD*, *bacA*, *baeS*, and *dfnA*) possessed by strain SBB were analyzed by RT-qPCR using cDNA obtained by a 1:5 dilution of reverse transcription as a template. The *Bacillus* gene *YvzC* was used as an internal reference gene due to its more stable expression level [45]. The reaction conditions were 94 °C for 5 min and 40 cycles of 95 °C for 10 s, 60 °C for 30 s, and 60 °C for 30 s. The reactions were performed on an ABI 7500 (Applied Biosystems, Carlsbad, CA, USA), and the relative expression of genes was calculated using the 2^−ΔΔCT^ method [46]. The experiment was repeated three times, and each treatment was conducted in triplicate. The primer sequences used are listed in Table 2.

### 2.11. Statistical Analysis

The data were treated using analysis of variance and Duncan’s multiple comparison or Independent Samples *t*-test to determine significant differences (*p* < 0.05) with SPSS 26.0 software (IBM Inc., Armonk, NY, USA). Graphs were drawn using GraphPad Prism 8.0 (GraphPad Software, Inc., San Diego, CA, USA).

## 3. Results

### 3.1. Identification of Strain SBB

The sequencing results were compared in the GenBank database, and all the results showed sequence homology between SBB and the corresponding genes of *B. velezensis* (>99%). The phylogenetic tree was constructed using the NJ method, and the 16S rRNA-based phylogenetic tree showed that strains SBB and *B. velezensis* RA-9 clustered together with a confidence level of 99% (Figure 1a). The phylogenetic tree based on the *gyrA* gene showed that strain SBB clustered with *B. velezensis* SBG9, with a confidence level of 100% (Figure 1b), and the phylogenetic tree based on *gyrB* showed that strain SBB clustered with *B. velezensis* FJ17-4, with a confidence level of 100% (Figure 1c). Thus, we identified *Bacillus* sp. SBB as *B. velezensis*.

### 3.2. Inhibition of V. dahliae Growth by SBB, Its VOCs, and Filtrate

A plate confrontation experiment showed that *V. dahliae* growth was significantly inhibited by strain SBB. The control colonies grew well at 28 °C for 10 d and covered almost the whole medium; in comparison, the colony diameter of *V. dahliae* treated with strain SBB was significantly inhibited (Figure 2a).

The result of the two sealed Petri dish experiments showed that the VOCs of strain SBB had a good inhibitory effect on *V. dahliae* growth, and after 10 d of incubation at 28 °C, the colony diameter of *V. dahliae* exposed to strain SBB VOCs treatment was significantly inhibited compared with the control group (Figure 2a).

The growth of *V. dahliae* treated with sterile fermentation filtrate of strain SBB at different times showed an obvious antagonistic effect on *V. dahliae* (Figure 2b). Calculation indicated that the 48-h fermentation filtrate had the best antagonistic effect on *V. dahliae*, with a 70.94% inhibition rate (Figure 2c).

### 3.3. Inhibition of V. dahliae Spore Production by SBB Filtrate

The experimental results showed that the medium of control and 1% concentration SBB filtrate treated were turbid, which indicated that a large number of spores were produced. The medium was clearer after treatment with 5%, 10%, and 20% concentration of SBB filtrate (Figure 3a), suggesting that the number of spores in these treatment groups was low. The spore solution of each treatment group was collected and counting and showed that the spore concentration in the control was approximately 4.5 × 10^7^/mL. The number of *V. dahliae* spores produced in the spore-producing medium with 5%, 10%, and 20% SBB filtrate was significantly lower, with the lowest for the 20% SBB filtrate group of approximately 1.3 × 10^6^/mL (Figure 3b), which was only 2.89% of the control group, indicating that SBB fermentation filtrate could have an antagonistic effect on *V. dahliae* by inhibiting its spore production.

### 3.4. Effect of SBB Filtrate on V. dahliae Microsclerotia and Melanin Formation

Under normal conditions, the microsclerotia produced by *V. dahliae* after 14 d of culture filled the field of view, and the number of microsclerotia was significantly reduced after treatment with 1% SBB filtrate. The mycelia remained white, and no microsclerotia were produced after treatment with 5%, 10%, and 20% SBB filtrate (Figure 4a). The density of microsclerotia for the control medium was approximately 29/mm^2^, 8/mm^2^ for the 1% treatment, and 0 for the 5%, 10%, and 20% treatments (Figure 4b). This result indicates that SBB fermentation filtrate inhibited the ability of *V. dahliae* to produce microsclerotia.

The melanin produced by *V. dahliae* in the control group was approximately 0.07 g/L, and that following the treatment with strain SBB was approximately 0.043 g/L (Figure 4c). Compared with the control group, the melanin content was significantly reduced by 44.28%, indicating that strain SBB effectively inhibited *V. dahliae* melanin production.

### 3.5. Identification of Genes Encoding Antimicrobial Substances of SBB

The results of PCR amplification using the genomic DNA of strain SBB as a template for genes related to regulatory processes or encoding antimicrobial substances that may be contained in strain SBB are shown in Figure 5. The amplified band lengths were approximately 1600, 1600, 1600, 1300, 1047, 594, 1203, 1200, and 2000 bp for genes *fenB*, *fenD*, *srfAA*, *srfAD*, *ituA*, *ituC*, *ituD*, *bmyA*, and *mycB,* respectively, encoding five of the non-ribosomal lipopeptides fengycin, surfactin, iturin, bacillomycin, and mycosubtilin, consistent with the expected band sizes. The amplified band sizes of *bacA*, *bacD*, and *bacAB*, encoding one small-molecule peptide bacilysin, were approximately 1200, 815, and 500 bp, respectively, consistent with expected sizes. The length of the amplified band of *dhbA*, the gene encoding the siderophore bacillibactin, was approximately 1350 bp, similar to the expected size. Genes *baeS* and *dfnA*, encoding the two polyketides bacillaene and difficidin, amplified two specific bands of approximately 1550 and 1950 bp, respectively, consistent with the expected size. The above results indicated that strain SBB may produce at least nine different antifungal substances.

### 3.6. Detection of Strain SBB Extracellular Enzyme Siderophore Production

Strain SBB was inoculated onto six different extracellular enzyme assay media, and the results showed that SBB produced clear circles on amylase, cellulase, and protease assay media (Figure 6), but not for phosphodiesterase, chitinase, and β-1,3-glucanase. Thus, SBB produced amylase, cellulase, and protease but not phosphodiesterase, chitinase, and β-1,3-glucanase during growth.

The CAS plate showed a yellow halo around it after the addition of 50 μL of the fermentation filtrate of ferritin (Figure 6d), indicating that SBB produced siderophore during its growth.

### 3.7. GC–MS Analysis of VOCs Produced by SBB and Antifungal Activity of Commercial VOCs against V. dahliae

A total of nine compounds with relative peak areas greater than 0.5% were obtained after removing the same volatiles produced by LB medium without strain SBB (Table 3). The dominant VOC detected in this study was 2-heptanone, with a peak area of 14.92% at 5.25 min, followed by 6-methyl-2-heptanone, with 9.45% at 6.19 min, and 2-nonanone, with 8.51% at 8.71 min (Figure 7).

To demonstrate their potential antifungal activity, we purchased standards (Table 3), among which 2-decanone and 2-tetradecanone had no antifungal effect on *V. dahliae*. The other seven standards were diluted to different concentrations and co-cultured with *V. dahliae* to observe their antagonistic effects (Figure 8a). In the concentration range of 0.06–1.3 μL/mL, all seven standards were volatile and inhibited mycelial growth. Among them, 2-nonanol (0.06 μL/mL), 2-heptanone (0.3 μL/mL), 6-methyl-2-heptanone (0.3 μL/mL), and 2-nonanone (0.6 μL/mL) completely inhibited *V. dahliae* growth. The 2-undecanone at higher concentrations (0.3 μL/mL) inhibited growth by more than 50%, while the inhibitory effect of 2-dodecanone and 2-tridecanone was weak, and even the high concentration (1.3 μL/mL) did not greatly inhibit *V. dahliae* mycelial growth (Figure 8b).

### 3.8. Gene Expression Analysis of Antifungal Substances in the Antagonistic Process of SBB against V. dahliae

In order to better search for key inhibitory substances of SBB antagonizing *V. dahliae*, we analyzed the differences in expression of genes present in SBB that regulate or encode the production of inhibitory substances using RT-qPCR. Compared with the control, the genes *ituC* and *bmyA* encoding the non-ribosomal lipopeptides iturin and bacillomycin, the genes *baeS* and *dfnA* encoding the polyketides bacillaene and difficidin, and genes *bacD* and *bacAB* encoding the small-molecule peptide bacilysin were all upregulated, with *dfnA* expression the most upregulated at 1.923-fold, followed by *bacAB* at 1.848-fold, and *baeS*, *bacD*, *bmyA*, and *ituC* at 1.448-, 1.364-, 1.236-, and 1.197-fold, respectively. No differential changes were observed in the expression of genes *ituD*, *srfAD*, and *mycB*, encoding non-ribosomal lipopeptides iturin, fengycin, and mycosubtilin, and of *bacA*, encoding the small-molecule peptide bacilysin (Figure 9).

## 4. Discussion

The first target gene considered for bacterial identification was 16S rRNA; however, when 16S rRNA was used to identify strains of the amylolytic family, the results were not satisfactory, and there is a precedent of *B. velezensis* being classified as amylolytic [47]. With the development of biotechnology, many functional genes that are very conserved during phylogeny have been used for specific analysis and are more effective in distinguishing *Bacillus* species relative to 16S rRNA genes, such as *gyrA*, *cheA*, *gyrB*, *rpoA*, *rpoB*, *vrrA*, and *spoOA*. In order to better define the isolation status of SBB, this study further cloned three specific genes (16S rRNA, *gyrA*, and *gyrB*), and also constructed a phylogenetic tree. This resulted in the accurate identification of SBB as *B. velezensis* and lay the foundation for subsequent research and application of this strain.

*Bacillus*, as commonly used biosynthetic bacteria, have a good effect in controlling plant fungal diseases. In this study, *B. velezensis* had an extremely strong inhibitory effect on *V. dahliae* growth in plate standoff experiments. Subsequently, it was confirmed that the VOCs produced by SBB also had a good inhibitory effect on *V. dahliae*, with 56.19% inhibition by Petri dish-to-button fumigation. The experimental study of the inhibitory effect of the SBB sterile fermentation filtrate at different time periods on *V. dahliae* showed that this filtrate at all time periods had a good inhibitory effect on *V. dahliae* growth, with the effect at 48 h being the most prominent. This series of experimental results showed that SBB and its VOCs and sterile fermentation filtrate all had good inhibitory effects on *V. dahliae*. The result that the 48-h fermentation filtrate of SBB had the best inhibitory effect on *V. dahliae* is beneficial for subsequent extraction and utilization of the inhibitory substances, saves production time, and has good potential for exploitation compared to previous study on the best inhibitory effect of *B. velezensis* fermentation filtrate of 72 h [48].

*Verticillium dahliae* reproduces and grows mainly by spore germination and forms microsclerotia in response to adversity. It has been demonstrated that the aggressiveness of *V. dahliae* correlates with the rate of spore production in plants, with high spore germination rate *V. dahliae* being more aggressive. Each individual cell of microsclerotia has the ability to germinate once, which increases the chance of establishing a successful infection and makes microsclerotia important targets for disease control [1,49]. Melanin could provide protection to the micronuclei and may be correlated with the virulence in *V. dahliae* [50,51]. Considering these characteristics, this experiment investigated the inhibitory mechanism on *V. dahliae* by measuring the effects of strain SBB on spore production, micronucleus production, and melanin production of *V. dahliae*. The *V. dahliae* spore production was significantly inhibited by the addition of 5% SBB sterile fermentation filtrate. In addition, in the experiment on the effect on *V. dahliae* micronucleus production, the addition of 5% SBB filtrate prevented *V. dahliae* from producing micronuclei. Experimental results showed that the melanin content in the mycelium of *V. dahliae* treated with SBB was approximately 0.043 g/L, which was 61.43% of the control group value (0.07 g/L). The results showed that SBB could antagonize *V. dahliae* by inhibiting mycelial growth, reducing spore production and slowing the growth of *V. dahliae* by inhibiting the formation of micronuclei, thus reducing its ability to survive adversity and reducing melanin production and so showing the potential ability to limit the pathogenicity of *V. dahliae*. This result is consistent with previous reports that the sterile fermentation filtrate of *B. tequilensis* C-9 inhibited the micronucleus formation of *V. dahliae* [52].

Previously, it was reported that *Bacillus* could act as a fungal inhibitor through the production of lipopeptides, various extracellular enzymes and protein-like substances, and VOCs [53,54,55]. We investigated the various types of antimicrobial substances that may be produced by strain SBB. First, we inoculated SBB on various extracellular enzyme assay media and showed that SBB could produce three extracellular enzymes during growth: cellulase, amylase, and protease. Cellulose is an important component of the fungal cell wall, and the cellulase produced by SBB inhibits the ability of the fungus to produce cellulose, impeding synthesis of the fungal cell wall as well as destroying it [56]. Glycoproteins and ribosomal proteins in the *V. dahliae* cell wall are inhibited by proteases produced by strain SBB [57]. The amylase produced by SBB can target and reduce the extracellular polysaccharides secreted by, and weaken antioxidant capacity of, the pathogen [58]. In addition, SBB was found to produce siderophore, which means it may reduce the availability of iron for the survival of *V. dahliae* through siderophore-mediated competition and finally inhibit *V. dahliae* growth [59].

Previous researchers found that VOCs produced by bacteria could be a mechanism of biocontrol against some soilborne fungal diseases [60]. Moreover, VOCs are regarded as environmentally friendly biological compounds as they can easily evaporate at normal temperature and pressure and diffuse through atmosphere and soil over short and long distances, are biodegradable, and do not leave toxic residues on plant surfaces [61]. For the production of volatile gases by SBB, we determined their composition using headspace solid-phase microextraction combined with GC-MS, and seven antimicrobial substances were analyzed, among which 2-nonanol (0.06 μL/mL), 2-heptanone (0.3 μL/mL), 6-methyl-2-heptanone (0.3 μL/mL), and 2-nonanone (0.6 μL/mL) could completely inhibit *V. dahliae* growth. Lee et al. [62] extracted 2-heptanone and 6-methyl-2-heptanone from the volatile gas of *B. amyloliquefaciens* DA12 and confirmed the strong inhibitory effect of these two substances on fungus of the genus *Fusarium*. In addition, Ren et al. [63] extracted 2-nonanol from the volatile substances produced by *B. velezensis* ZJ and confirmed its strong inhibitory effect against *Botrytis cinerea*. However, there is no research report on the antifungal activity of 2-nonanone, and our results suggested for the first time that 2-nonanone had a strong inhibitory effect on *V. dahliae*, and also provided reference for the future development and utilization of chemical fumigation agents.

We performed RT-qPCR on the genes related to the synthesis of antifungal substances in strain SBB in order to clarify the substances that play a major role in the inhibition of *V. dahliae* by SBB. The results showed that the genes *ituC* and *bmyA*, encoding non-ribosomal lipopeptides iturin and bacillomycin, genes *baeS* and *dfnA*, encoding polyketides bacillaene and difficidin, and genes *bacD* and *bacAB*, encoding the small-molecule peptide bacilysin were, upregulated during the inhibition of *V. dahliae* by SBB. Therefore, when SBB inhibits *V. dahliae*, the dominant antimicrobial substances are likely bacillomycin, bacillaene, difficidin, and bacilysin, which showed significantly upregulated expression of the corresponding encoding genes. The next step should be to isolate and purify these substances, or knock out specific genes, to investigate the antifungal mechanism of SBB at the molecular level.

## 5. Conclusions

*Bacillus velezensis* SBB significantly inhibited the sporulation, microsclerotia, and melanin formation of *V. dahliae*. The mixed VOCs released by SBB and the corresponding commercial standards decreased *V. dahliae* mycelial growth. During the antagonism process, amylase, protease, cellulase, and siderophore were produced, and there was significantly upregulated expression of genes encoding four antifungal substances: bacillomycin, bacillaene, difficidin, and bacilysin. Therefore, SBB has the potential to be exploited as a biopesticide against Verticillium wilt.

## Figures and Tables

**Figure 1 jof-08-01021-f001:**
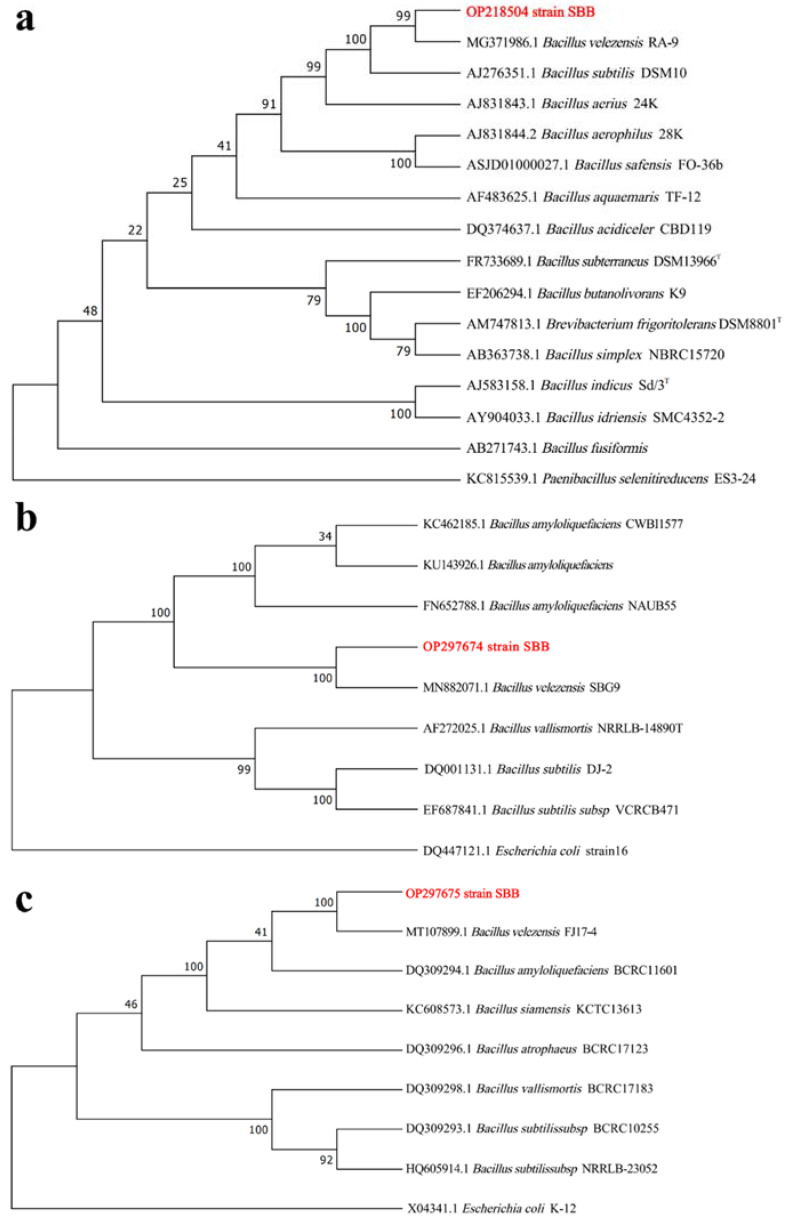
Phylogenetic tree of *Bacillus* sp. SBB based on (**a**) 16S rRNA, (**b**) *gyrA*, and (**c**) *gyrB* gene sequences. *Paenibacillus selenitireducens* ES3-24, *Escherichia coli* K-12, and *E. coli* strain 16 were selected as outgroups for the 16S rRNA, *gyrA*, and *gyrB* phylogenetic trees, respectively.

**Figure 2 jof-08-01021-f002:**
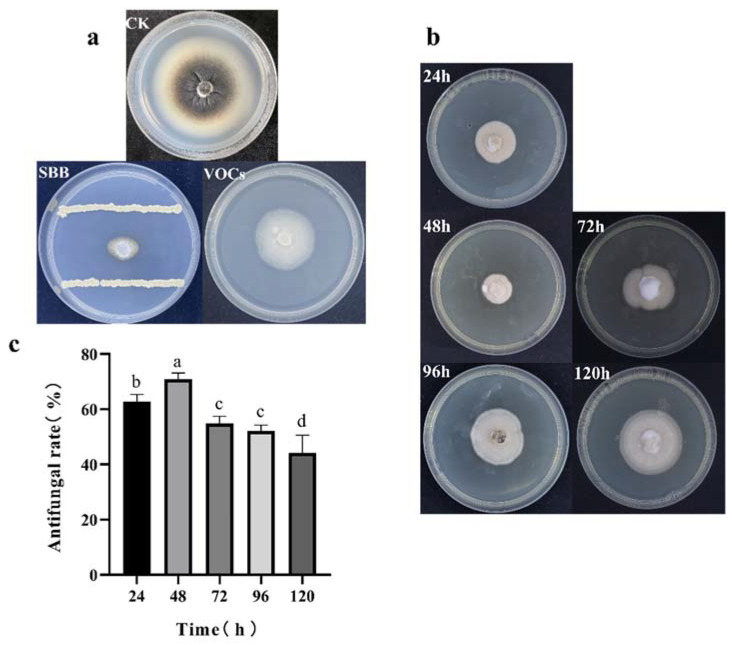
Inhibition of *Verticillium dahliae* by (**a**) strain SBB and its VOCs, (**b**) aseptic fermentation filtrate of SBB at different periods, and (**c**) the antifungal rate of aseptic fermentation filtrate of SBB. CK: fungi that were not inoculated with bacteria served as a control. The data were analyzed by one-way ANOVA followed by Duncan’s post-hoc test. Different letters indicate statistically significant differences (*p* < 0.05) among treatments.

**Figure 3 jof-08-01021-f003:**
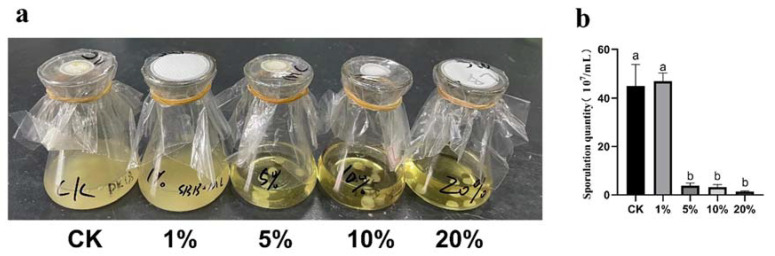
Effect of fermentation filtrate of strain SBB on sporulation quantity of *Verticillium dahliae*: (**a**) sporulation media with different treatments and (**b**) quantitative analysis of *V. dahliae* sporulation. The data were analyzed by one-way ANOVA followed by Duncan’s post-hoc test. Different letters indicate statistically significant differences (*p* < 0.05) among treatments.

**Figure 4 jof-08-01021-f004:**
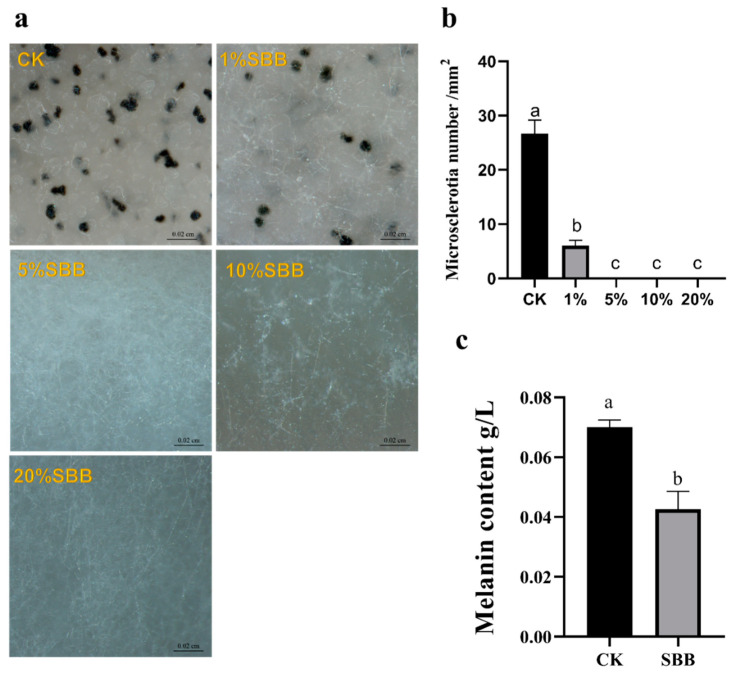
Effect of strain SBB fermentation filtrate on the formation of microsclerotia and melanin of *Verticillium dahlia*: (**a**) production of microsclerotia in different treatment groups under the microscope; (**b**) quantitative analysis of the microsclerotia of *V. dahliae*, scale = 0.02 cm; and (**c**) quantitative analysis of the melanin content of *V. dahliae*. The data were analyzed by one-way ANOVA followed by Duncan’s post-hoc test. Different letters indicate statistically significant differences (*p* < 0.05) among treatments.

**Figure 5 jof-08-01021-f005:**
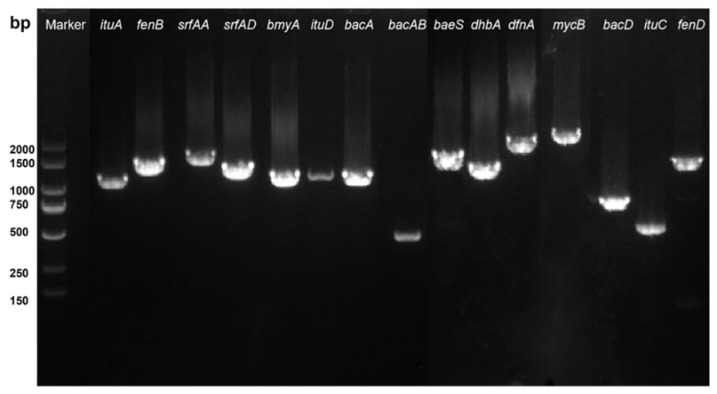
PCR detection of 15 biosynthetic genes (*ituA*, *fenB*, *srfAA*, *srfAD*, *ituD*, *bacA*, *bacAB*, *baeS*, *dhbA*, *mycB*, *bacD*, *ituC*, and *fenD*) of antifungal substances in *Bacillus velezensis* SBB.

**Figure 6 jof-08-01021-f006:**
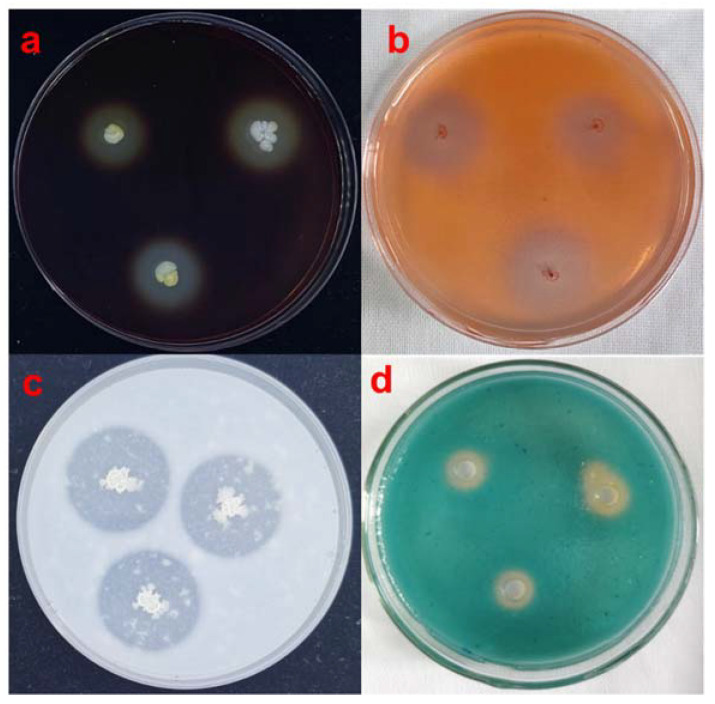
Detection of (**a**) amylase, (**b**) cellulase, (**c**) protease, and (**d**) siderophore activity of *Bacillus velezensis* SBB.

**Figure 7 jof-08-01021-f007:**
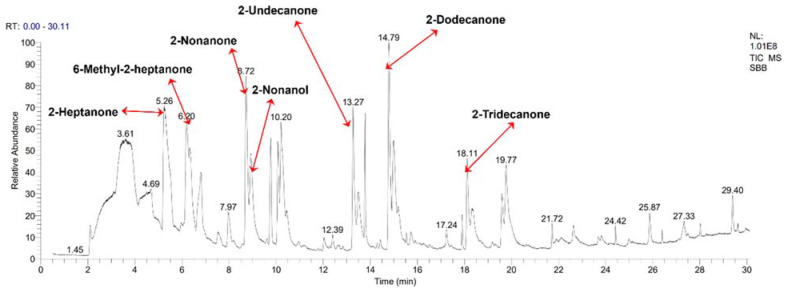
Gas chromatograph-mass spectrometer analysis of volatile organic compounds in *Bacillus velezensis* SBB.

**Figure 8 jof-08-01021-f008:**
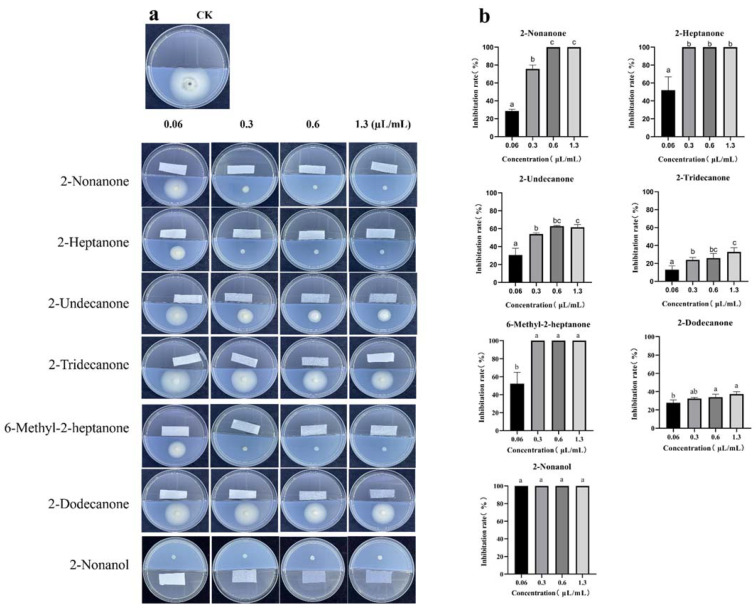
(**a**) Inhibitory effect of standard substances with different volumes (0.06, 0.3, 0.6, and 1.3 µL/mL) on *Verticillium dahliae* and (**b**) inhibition rates of different standard substances against *V. dahliae*. The data were analyzed by one-way ANOVA followed by Duncan’s post-hoc test. Different letters indicate statistically significant differences (*p* < 0.05) among treatments.

**Figure 9 jof-08-01021-f009:**
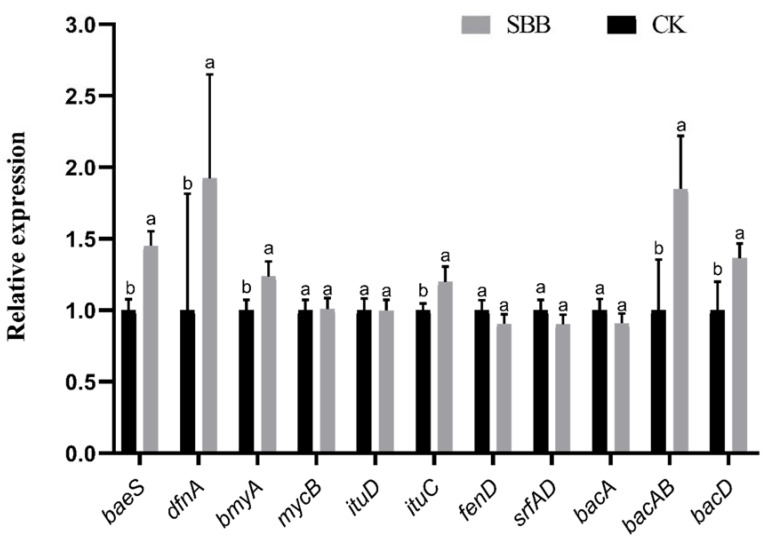
Relative expression levels of *Bacillus velezensis* SBB antibiotic-related genes. Data are the means ± SE. Different letters indicate statistically significant differences (*p* < 0.05) among treatments (*t*-test).

**Table 1 jof-08-01021-t001:** Primers for gene amplification related to antifungal substances in strain SBB.

Lipopeptides	Gene	Primers	Sequences (5′–3′)	PCR Products Size (bp)	Reference
Surfactin	*srfAA*	srfAA-F	GCCCGTGAGCCGAATGGATAAG	1600	[34]
		srfAA-R	CCGTTTCAGGGACACAAGCTCCG		
	*srfAD*	srfAD-F	CCGTTCGCAGGAGGCTATTCC	1300	[34]
		srfAD-R	CCGTTCGCAGGAGGCTATTCC		
Fengycin	*fenB*	fenB-F	CTATAGTTTGTTGACGGCTC	1600	[35]
		fenB-R	CAGCACTGGTTCTTGTCGCA		
	*fenD*	fenD-F	TTTGGCAGCAGGAGAAGTTT	1600	[36]
		fenD-R	GACAGTGCTGCCTGATGAAA		
	*ituA*	ituA-F	ATGTATACCAGTCAATTCC	1047	[37]
		ituA-R	GATCCGAAGCTGACAATAG		
Iturin	*ituC*	ituC-F	CCCCCTCGGTCAA GTGAATA	594	[37]
		ituC-R	TTGGTTAAGCCCTGATGCTC		
	*ituD*	ituD-F	ATGAACAATCTTGCCTTTTTA	1203	[35]
		ituD-R	TTATTTTAAAATCCGCAATT		
Bacillomycin	*bmyA*	bmyA-F	AAAGCGGCTCAAGAAGCGAAACCC	1200	[37]
		bmyA-R	CGATTCAGCTCATCGACCAGGTAGGC		
Mycosubtilin	*mycB*	mycB-F	ATGTCGGTGTTTAAAAATCAAGTAACG	2000	[38]
		mycB-R	TTAGGACGCCAGCAGTTCTTCTATTGA		
	*bacAB*	bacAB-F	CAGCTCATGGGAATGCTTTT	500	[38]
		bacAB-R	CTCGGTCCTGAAGGGACAAG		
Bacilysin	*bacD*	bacD-F	CTTCTCCAAGGGGTGAACAG	815	[38]
		bacD-R	TGTAGGTTTCACCGGCTTTC		
	*bacA*	bacA-F	GTGAAGGCCGTACTTTTGTCTGGC	1200	[34]
		bacA-R	GGGGGGAAATACAGCTTCAGGGC		
Bacillaene	*baeS*	baeS-F	CGCAAAAGCTCTTCGACCGCCGTC	1550	[34]
		baeS-R	CTCTCGTGCCGTCGGAATATCCGC		
Difficidin	*dfnA*	dfnA-F	GGTGCGGCATGAAGATTTGAGATCACCG	1950	[34]
		dfnA-R	GGAGAGCACTTCAATTCCGACGTTGACC		
Bacillibactin	*dhbA*	dhbA-F	CGCCTAAAGTAGCGCCGCCATCAACGC	1350	[34]
		dhbA-R	CCGCGATGGAGCGGGATTATCCG		

**Table 2 jof-08-01021-t002:** Primers of genes related to antifungal substances in strain SBB for RT-qPCR.

Gene	Primers	Sequences (5′–3′)	Function
*YvzC*	qYvzC-F	GTGGTCGAAAAGATACA	Internal reference gene
	qYvzC-R	TCATTGACTTTTGTCAAC	
*srfAD*	qsrfAD-F	AATGCTGTCTGTCATATCCTATA	Regulating the synthesis of surfactin
	qsrfAD-R	GTGGCGGAACGAATCTAT	
*fenD*	qfenD-F	TCATTACCGACTCCATCA	Regulating the synthesis of fengycin
	qfenD-R	TTCTGTTTCCTGTGTTCAA	
*ituC*	qituC-F	CGGAGACACATACACTTC	Regulating the synthesis of iturin
	qituC-R	GTTCGTTTCATCTGTTCTTC	
*ituD*	qituD-F	GAATCACATTGTACCTTATC	Regulating the synthesis of iturin
	qituD-R	CGTCGTCATATTGGAATA	
*bmyA*	qbmyA-F	CGATTTCCTTCCAATACG	Regulating the synthesis of bacillomycin
	qbmyA-R	AACAATATACGGACAACAC	
*mycB*	qmycB-F	AATTGAACGCTGGTCTAA	Regulating the synthesis of mycosubtilin
	qmycB-R	AATGCTGAAGGTGAAGTC	
*bacAB*	qbacAB-F	ATGTCATCTGTATATTCAA	Regulating the synthesis of bacilysin
	qbacAB-R	CATTAAGCACTTCTACAT	
*bacD*	qbacD-F	GATAACGGAGTAAGACAATA	Regulating the synthesis of bacilysin
	qbacD-R	GACTTCCTTATGCTGATG	
*bacA*	qbacA-F	CAAGGCTTATTCAATATCTA	Regulating the synthesis of bacilysin
	qbacA-R	CACGATTCAAATGTATCA	
*baeS*	qbaeS-F	AACGCATTCATTCACATC	Regulating the synthesis of bacillaene
	qbaeS-R	ACAACGGCTCATAAGTAT	
*dfnA*	qdfnA-F	ATAACGCATTATATTCTC	Regulating the synthesis of difficidin
	qdfnA-R	CATATTAGGCTACTCTAT	

**Table 3 jof-08-01021-t003:** GC–MS/MS VOC profile of *B**acillus*
*velezensis* SBB.

Retention Time (min)	Relative Peak Area (%)	CAS#	Compound
5.25	14.92	110-43-0	2-Heptanone
6.19	9.45	928-68-7	6-Methyl-2-heptanone
8.71	8.51	821-55-6	2-Nonanone
8.93	2.76	628-99-9	2-Nonanol
10.07	2.63	693-54-9	2-Decanone
13.27	4.89	112-12-9	2-Undecanone
14.79	7.47	6175-49-1	2-Dodecanone
18.11	3.22	593-08-8	2-Tridecanone
19.59	1.66	2345-27-9	2-Tetradecanone

## Data Availability

All data generated or analyzed during this study are included in this article.

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
