# Peer review of "Identification of Bacillus velezensis SBB and Its Antifungal Effects against Verticillium dahliae"

_jof, 2022, doi:10.3390/jof8101021_

Round 1
Reviewer 1 Report
Dear authors
Your article is very interesting but you should revise it before resubmit again to this Journal
extensive reading by English-native reader is necessary for a suitable publication in this international journal
it is better to added good recommendation in abstract
Also write your objective clear
put the names of the author(s) of all taxa cited the first time they appear in the text
2.1. Fungal Strains and Growth Media this title about fungi but you wrote about bacteria
Figure 5. PCR detection of biosynthetic genes of antifungal substances in Bacillus velezensis SBB.
this Fig should be more explain
conclusion should rewrite again and focus on your best results
Author Response
Dear reviewer,
Thank you for your haplful comments. Please see the attachment for the response to the comments.

Reviewer 2 Report
The work conducted on the inhibitory effect of B. velezensis strain SBB on Verticillium dahliae adequately describes the interaction of the two microorganisms in vitro. Overall, the work will be acceptable for publication after some revisions.

Author Response

(The authors gave the same response as above.)
